# Anesthetics and Long Term Cancer Outcomes: May Epigenetics Be the Key for Pancreatic Cancer?

**DOI:** 10.3390/medicina58081102

**Published:** 2022-08-14

**Authors:** Zhirajr Mokini, Alessandro Cama, Patrice Forget

**Affiliations:** 1ESAIC Mentorship Program, BE-1000 Brussels, Belgium; 2The European Platform for Research Outcomes after PerIoperative Interventions in Surgery for Cancer Research Group (Euro-Periscope): The Onco-Anaesthesiology Research Group (RG), BE-1000 Brussels, Belgium; 3Department of Pharmacy, G. d’Annunzio University of Chieti-Pescara, 66100 Chieti, Italy; 4Epidemiology Group, Institute of Applied Health Sciences, School of Medicine, Medical Sciences and Nutrition, University of Aberdeen, Aberdeen AB25 2ZD, UK; 5Department of Anaesthesia, National Health Service (NHS) Grampian, Aberdeen AB25 2ZD, UK

**Keywords:** cancer, anesthesia, epigenetics, ketorolac, lidocaine, propofol, sevoflurane, propranolol, repurposing drugs, miRNA, pancreatic cancer

## Abstract

Knowledge shows a divergence of results between preclinical and clinical studies regarding anesthesia and postoperative progression of cancer. While laboratory and animal data from then 2000s onwards raised much enthusiasm in this field of research leading to several clinical investigations worldwide, data from randomized trials seem to have killed off hope for many scientists. However several aspects of the actual knowledge should be reevaluated and there is space for new strategies of investigation. In this paper, we perform a critical review of actual knowledge and propose new research strategies with a special focus on anesthetic management and repurposed anesthetic adjuvants for pancreatic cancer.

## 1. Introduction

Despite considerable efforts, pancreatic cancer (PC) is one of the deadliest cancers and its incidence and mortality is increasing. Exocrine pancreatic ductal adenocarcinomas (PDAC) represent 90–95% of PCs, whereas the remaining 5% arise from the endocrine pancreas. The 5-year survival rate for all stages is 7–11% and only 42% in people with local disease (13%). Surgery remains the best treatment option but resectable cancers represent less than 25% of all cases and only multidisciplinary approaches have shown to be able to improve cure. The development of new drugs is a long, expensive and very challenging process. For these reasons, there is a renewed interest in research towards perioperative and repurposed interventions that may improve mortality after PC. The main strengths of repurposed drugs is the availability of a well-documented clinical use and pharmacological and safety data, the possibility to start from phase II clinical trials to assess their efficacy for the new indication, and the reduced cost [1].

Starting from late the 1970s, experimental and clinical data have suggested that standard anesthetic techniques for cancer surgery, general or regional anesthesia, act differently in cancer biology and may affect its progression-related outcomes [2,3].

During the perioperative period, anesthetics can act either on cancer cells or through modifications in host response. Modifications in host response occur both systemically and locally on the surgical site [2,3]. Regional anesthesia prevents the stress and immune response that follows surgery whereas general anesthesia does not, thus favoring tumor growth in basic research studies. Alterations in cell immunity have been observed up to a week after surgery and are believed to promote local residual micro-metastases as well as a favorable local microenvironment and systemic situation for the invasion of circulating tumor cells [4,5].

However, the situation is extremely complex. This is evident in true life, where basic research findings led to the conduction of several clinical trials worldwide investigating the repurposed action of anesthetics among other drugs [1]. These trials have been mostly negative and this finding is confirmed by meta-analyses [6,7,8,9], but again, to date, there is no clear response as to whether it is the extent of surgery rather than anesthesia management that most influences cancer outcomes. Furthermore, there is some concern, since some negative trial results can be due to high surgical curability that may mask the effects of anesthesia. Factors such as tumor characteristics, extent of surgical damage and the related stress response and surgical curability contribute to determine the long term desired clinical results and should be better taken into account in the design of clinical trials.

Several aspects of the puzzle have not yet been adequately explored and further and new research strategies are needed to explore the basic mechanisms by which anesthesia acts on cancer. The question that remains is which biological mechanism and anesthetic management, lasting a few hours or days, could influence the natural history of cancer changing outcomes for months or years after surgery? In this review, the general aspects of anesthetic role in epigenetics are described with a special focus on pancreatic cancer.

## 2. Epigenetics

The biological response of cells under the influence of physiological and external factors is driven by epigenetic modulation of gene and protein expression. To make an analogy, genes are the constitution and epigenetic changes are all genetic mechanisms other than DNA sequence changes. Epigenetics are laws, regulatory acts and amendments, or the heritable cellular system that interprets the genome [10,11,12,13,14].

In normal cells, the transcriptional status of most genes, the coding part of the genome, is epigenetically fixed through the binding of DNA with histones. However, other genes reside in a balanced state sensitive to exogenous signals and capable of rapid modulation of DNA transcription to mRNA. These events lead to alterations in gene expression, a consequent differential production of messenger RNAs (mRNAs) and a subsequent modified expression of proteins that are the final biologic effectors. Several short- or long-time-operating factors including external chemical or pharmacological stimuli can alter this balance [15,16]. One of the most significant examples of the effect of epigenetics is synaptic plasticity.

Anesthetics are chemical factors with the potential to induce epigenetic effects. With regard to the nervous system, for example, this is supported by the fact that anesthetics are neurotoxins sharing many molecular mechanisms of action with alcohol and cocaine [17,18]. The effect on epigenetics occurs even after relatively short exposures, but is more intense after prolonged or repeated exposures [19]. Most importantly, the neurocognitive effects associated with epigenetic changes after anesthetic exposure may persist in neonates or be present for weeks to years in the elderly [18]. Moreover, anesthetic exposure may affect not only the exposed subjects, but also future generations [18].

However epigenetics is a relatively new branch of genetics which only began to be investigated 50 years ago. Research on epigenetics and anesthesia is even younger and our understanding of the biological effects of anesthetics and the underlying molecular mechanisms is uncomplete [18].

### 2.1. Epigenetics and Anesthetics

Research on the epigenetic effects of anesthetics has been mainly focused on the neurodevelopmental brain alterations and ischemia reperfusion injury effects. The first microarray investigation on inhalational anesthetics showed that expression of 1.5% of 10,000 genes in various organs was altered [20]. Sevoflurane has the ability to change the expression of the circadian genes and drug metabolizing enzymes [21,22]. Propofol and sevoflurane show a different protein expression change activity in rat brains [23]. The effects are more pronounced at young ages, when the central nervous system and other tissues are highly susceptible to what is called epigenetic reprogramming [24,25,26,27]. Anesthetics are also capable of inducing prolonged and intergenerational epigenetic effects. One week after bariatric surgery under general anesthesia, DNA methylation of 1509 genes in male spermatozoa remains altered and 1004 of those genes remained altered after 1 year [28]. Reduced DNA methylation in the 5-upstream promoter region of rat mothers exposed to 6 h sevoflurane and upregulation of Arc and JunB mRNA expression, two genes regulating synaptic plasticity and neuronal development, are trans-generationally expressed in offspring male born [29].

### 2.2. Post-Translational Chromatin Regulation

Post-translational epigenetic modifications on histones and chromatin are regulated by more than 700 enzymes categorized as writers/erasers, readers, movers shapers and insulators. Lysine-rich N-terminal histone “tails” undergo several processes such as acetylation and methylation, ubiquitination, phosphorylation, and sumoylation [30].

Histone acetylation by acetyltransferases allows chromatin liberation and gene transcription, while histone deacetylation by deacetylases results in a stronger histone bond with DNA and inhibition of gene transcription [31]. Repeated sevoflurane exposure of neonatal rats leads to an increased hippocampal deacetylase activity and reduced histone acetylation, with developmental effects that are reversed by histone deacetylase inhibitors [32].

Methylation by DNA methyltransferases (DNMTs) can both induce and repress gene transcription, depending on the chromatin residue modified. Sevoflurane neurotoxicity on rat stem cells is exerted through a concentration-dependent DNA methylation [17]. There is also an increasingly growing evidence of the epigenetic effects of anesthetics on the developing human brain that include both methylation and acetylation [33].

DNA methylation is one of the mechanisms affected by stress and involved in postoperative hyperalgesia [34]. DNA methylation and DNMT expression in skin after incision is changed and controls nociceptive sensitization. DNMT inhibition attenuates incision-induced nociceptive hypersensitivity via up-regulation of Oprm1 gene expression while treatment with naloxone exacerbates incision-induced mechanical hypersensitivity [34].

DNA methylation in blood mononuclear cells after general anesthesia for major breast surgery is globally reduced by 26% while DNMT mRNAs expression is reduced by 65 to 71% [35]. Opiates are thought to increase methylation whereas lidocaine show controversial effects [36,37,38]. These findings support the evidence that anesthesia/surgery may alter the epigenetic status of host’s tissues after surgery.

Aberrant DNA methylation leads to onco-suppressor gene silencing or oncogene activation and has been linked to oncogenesis in a number of tumor types. Cancers present hypermethylation of CpG islands (DNA regions with a high density of cytosine–guanine dinucleotides) in or near promoter regions, whereas gene bodies are hypomethylated.

### 2.3. RNA

Only 1.5% of mammalian DNA encodes for approximately 20,000 genes which will be translated into proteins. Protein synthesis is driven by three main types of RNA: messenger RNA (mRNA), transfer RNA (tRNA) that accounts for the majority of RNA molecules, and ribosomal RNA (rRNA). mRNA is transcribed directly from DNA and contains the “message” that is translated by tRNAs into proteins. rRNA accounts for 90% of total RNA mass and forms ribosomes, which are the protein synthesizers [39]. The rest of the human genome, more than 90%, represents non-coding RNAs (ncRNA). The total number of RNA molecules is estimated at 107 per cell [40].

Non-coding RNAs are classified into long [>200 nucleotides] and short ncRNAs (<200 nucleotides) (lncRNA and sncRNA respectively). A series of biological functions such as regulation of the expression of coding genes, cancero-genesis and regulation of biological processes have been described for many ncRNAs, supporting the hypothesis that they are functional, although for most of them a biological role has not yet been discovered [41,42].

Micro RNAs (miRNA) are non-coding, endogenous and highly conserved sncRNAs across species. They act via post-transcriptional degradation or translational repression by binding to 3′ untranslated regions (UTR) of target mRNAs. Each molecule of mRNA can be regulated by more the one miRNA, and a single miRNA may influence the expression of a wide range of mRNAs [43]. Some miRNAs are expressed ubiquitously, whereas others are tissue-specific and/or stage-specific. miRNAs regulate the activity of 30–50% of protein-coding genes and modulate the expression of 10–30% of human genome [44,45]. RNAs also exist extracellularly in the circulation in exosomes [45].

Anesthetics affect miRNA in various cells across the body. In mice, sevoflurane and propofol affect expression of 46 of 177 miRNAs in liver, 20 miRNAs in lung, and 14 miRNAs in brain, with a specific pattern of expression at each anesthetic exposure [46,47,48]. Sevoflurane inhibits the NF-κB pathway through miRNA-9-5p expression and protects the liver from ischemia-reperfusion injury [49]. Sevoflurane also induces miRNA-155 downregulation that reduces systemic inflammation in acute lung injury models [50]. Propofol attenuates the neuroinflammation induced by lipopolysaccharide through miRNA-155 suppression [51].

Given these demonstrated effects on epigenetics, is it biologically plausible that anesthetics affect cancer’s natural history [18]? It is well known that epigenetic modulation in normal tissues may contrast or favor tumor progression, and anesthesia has the ability to change the epigenetics of surgical tissues [35,36,37,38,52]. Can anesthesia also directly alter cancer’s epigenetics? The question as to which mechanism can drive the anesthetic-induced alterations of tumor biology and host systems during a limited period of time such as the perioperative period, to the point that tumor behavior and its future clinical history change permanently, may have a response in epigenetics. However, to date there is no clear response a to whether it is the extent of surgery or stress response rather than anesthesia management that influences most cancer outcomes.

## 3. Anesthesia and Cancer Epigenetics

Epigenetics has a crucial role in cancero-genesis, which is a complex biological event. An altered gene and protein expression profile may affect cancer cells and the host response, inducing more or less favourable conditions for tumor progression. Cancer growth itself is driven by a “Darwinian” evolutionary process that starts in normal tissues where advantageous genetic and epigenetic events occur in a series of stages that promote clonal expansions of new tissue. Epigenetic changes can be subject to the same selection forces as genetic events [53]. This “new” tissue progressively misses the features of the original tissue. Every aspect of tumor biology including cell growth and differentiation, cell cycle control, cancer stem cell formation, DNA repair, angiogenesis, migration, and evasion of host immune surveillance is affected by epigenetic alterations [54,55].

Only few sporadic studies have investigated the role of anesthesia on cancer cell epigenetics and much is yet to be discovered (Figure 1). Differential gene expression was shown after ex-vivo exposure of brain cancer cell line SH-SY5Y and of breast cancer cell line MCF-7 to enflurane, isoflurane, desflurane, halothane, sevoflurane, and nitrous oxide [56].

Sevoflurane suppresses viability, invasion, migration, and apoptosis of colorectal cancer cells in a dose-dependent fashion by regulating the circ-HMGCS1/miRNA-34a-5p/SGPP1 axis, via inactivation of the Ras/Raf/MEK/ERK signaling, via regulation of ERK/MMP-9 pathway by up-regulating miRNA-203 and by regulating miRNA-34a/ADAM10 axis [57,58,59,60]. A study also showed a differential and specific impact on circulating exo-somial miRNAs during colon cancer surgery resection [61].

Other anesthetics such as the non-steroidal anti-inflammatory drugs, propofol and ketamine, may also modulate epigenetics. Celecoxib inhibits osteosarcoma cell proliferation, migration, and invasion via miRNA-34a [62]. miRNA-126-5p, −320a and -146a-5p regulate the sensitivity to celecoxib [63].

R-Ketorolac inhibits the activity of Rac1 and Cdc42, which are GTP-ases involved in cell growth and cell cycle regulation, in ovarian cancer patients. Both are believed to be therapeutic targets for ovarian cancer as regulators of migration, adhesion and invasion [64,65]. Studying epigenetics after ketorolac use may help in understanding the mechanism underlying this effect.

Lidocaine at clinical concentrations (1 mM) induced DNA demethylation for 120 h on BT-20 and MCF-7 breast cancer cells in vitro [66]. Lidocaine also reduces the proliferation of pancreatic cancer PaTu8988t cells after 48 h in vitro [67].

Ketamine, a N-methyl-Daspartate [NMDA] receptor antagonist used as a racemic derivative of two enantiomers s[+]ketamine and r[−]ketamine, can also modulate epigenetics. Exposure for 48 h with ketamine and s-ketamine of PaTu8988t pancreatic carcinoma cells significantly inhibited proliferation and expression of nuclear NFATc2 [67]. PaTu8988t and Panc-1 cells express the NMDA type R2a receptor. Ketamine and s-ketamine at 1000 μM concentration for 48 h significantly inhibited the proliferation of pancreatic cancer cells. S-ketamine reduced apoptosis after 3 h in PaTu8988t cells and in PANC-1 cells after 24 h incubation with ketamine [a] (65 ± 17%) and s-ketamine [b] (68 ± 24%). Necrosis increased after 16 h with ketamine and after 6–24 h after s-ketamine [68]. However, clinical doses of ketamine (2 mg/kg iv) produce an average plasma concentration of 41 μg/kg/min, which corresponds to a plasma concentration of 9.3 μM [68]. Ketamine has been recently found to suppress the viability of liver cancer cells and induce ferroptosis through the lncPVT1/miRNA-214-3p/GPX4 pathway [69]. In ovarian cancer, ketamine modulates the P300-mediated H3K27 acetylation activation in the promoter of lncRNA PVT1 that binds histone methyltransferase enhancer of zeste homolog 2 [EZH2], and regulates the expression of target genes, including p57, consequently regulating ovarian cancer cell growth, cell cycle control, apoptosis and colony forming [70].

Propofol (2,6-diisopropylphenol) is an extensively-used sedative anesthetic at 4-20 mcg/mL brain concentrations. Propofol regulates both miRNAs and lncRNAs, and modulates the signaling pathways of important oncogenes/onco-suppressors that are potential therapeutic targets including hypoxia-inducible factor-1α (HIF-1α), mitogen-activated protein kinase (MAPK), nuclear factor-kappaB (NF-κB), and nuclear factor E2-related factor-2 (Nrf2) [71,72].

Propofol upregulates miRNA-34a expression and induces miRNA-34a-dependent apoptosis in SH-SY5Y neuroblastoma cells in vitro [73,74]. At a concentration of 20 μg/mL for 72 h, propofol significantly reduces cell viability and apoptosis in human pancreatic PANC-1 cells increasing the expression of pro-apoptotic caspase-3 and Bax and down-regulating the expression of anti-apoptotic Bcl-2 gene. Apoptosis is induced by miRNA-3-4a-dependent upregulation of LOC285194 [75]. Additionally, propofol inhibits cell growth and metastasis by enhancing miR-328 expression in pancreatic cancer [76]. Propofol also induces a miRNA-34a-dependent E-cadherin upregulation in the PANC-1 cells, and reduces motility of cells is wound healing assays [75].

In human pancreatic Miapaca-2 and Panc-1 cells in vitro and murine pancreatic cancer cell (Panc02) in vivo, propofol shows a concentration-dependent (5, 25, 50, 100 μM) inhibition of cell migration, expression of VEGF and HIF-1α, phosphorylation of (ERK), AKT, (CaMK II), and Ca^2+^ concentration [77]. Propofol downregulates VEGF and suppresses migration of pancreatic cancer cells by inhibiting the NMDA receptor [77].

Propofol at concentrations of 1 to 10 μg/mL for 48 and 72 h also suppresses the proliferation and invasion of PANC-1 cells by 2.46- and 3.95-fold by upregulating miRNA-133a expression [78,79]. Apoptosis also increases after 1–10 microg/mL propofol exposure in a dose and time-dependent manner and through PUMA pathway [78,79].

Propofol inhibits the growth, invasion and migration of PANC-1 cells in a dose and time-dependent manner via the miRNA-21/Slug pathway and through E-cadherin upregulation [79]. Both PUMA and E-cadherin are upregulated by propofol via miRNA-21 inactivation and subsequent Slug inhibition [79].

The activation and expression of ADAM8 in response to hypoxia in PC is inhibited, thus antagonizing angiogenesis [80]. The downregulation of ADAM8 and upregulation of miRNA-328 mediated by propofol was shown to inhibit pancreatic cancer proliferation and metastasis [81]. Growth of xenograft pancreatic cancer is also inhibited in nude mice models by propofol [80]. Propofol shows a synergistic effect with gemcitabine through downregulation of NF-κB signaling pathway induced by gemcitabine, thereby promoting the chemosensitivity of PC [82].

## 4. Anesthetic Management

It remains uninvestigated how epigenetics changes after anesthesia and whether it may lead to an influence on cancer biology during the years after surgery, modifying its natural history and determining relevant clinical effects. To date, most available clinical studies are retrospective, small randomized trials or heterogeneous meta-analyses with negative or inconclusive results [6,7,8,9,83,84,85,86].

The major available trials have evaluated regional and general anesthesia showing that locoregional or epidural anesthesia do not affect cancer progression-related outcomes in breast (2132 patients), lung (400 patients) and cancer thoraco-abdominal surgery (1802 patients) [6,7,8]. These trials seem to definitively suggest that regional anesthesia-analgesia does not reduce recurrence after potentially curative cancer surgery [87,88].

However these trials were not constructed to detect differences between cancer subtypes or cancer grade. In one trial, only respectively did 13.5% and 12.5% of patients in the general anesthesia and epidural groups have advanced cancers [T3-4], whereas the other trial included multiple cancer types (gastrointestinal, hepatobiliary-pancreatic, lung-esophageal-thymic and genito-urinary and other cancers) [7,8].

The largest randomized controlled trial (The breast cancer recurrence trial, BCR), compared paravertebral to general anesthesia for breast cancer surgery. It included 2132 patients and showed no difference in cancer recurrence and survival between anesthetic techniques [6]. Contrarily to the study hypothesis based on residual disease, this trial included only low stage disease which has two implications. Firstly, surgery is highly curative at five years (>90%) in low grade disease as shown by the Mindact trial [89]. Secondly, surgery is these cases is less invasive leading to a lower stress response, lower pain and use of opioid, and reduced recurrence risk [90,91,92]. This is consistent with the finding that there is no metastatic progression after resection of primary breast cancer in murine orthotopic models if surgery is not associated with a large resection (laparotomy) [92]. In the human setting, patients having more invasive breast surgery (modified radical mastectomy), showed a significantly improved recurrence-free survival with propofol–total intravenous anesthesia compared with volatile-based anesthesia (HR, 0.55; 95% CI, 0.31 to 0.97; *p* = 0.037) [93].

60% of the participants of the BCR trial were enrolled from one hospital in China, which may have influenced the final results. Moreover, the study did not take into account surgical variability, was not able to detect patients with higher risk of recurrence, included the use of opiates for both groups, assessed only progesterone receptor status, and did not have an adequate follow-up time to assess recurrence in slow progression cancer phenotypes.

Based on the available preclinical and clinical data, amide local anesthetics and propofol are a plausible repurposing strategy frequently used on intravenous infusion for analgesia during cancer surgery. They are also a fundamental part in the setting of the opioid-free anesthesia strategy, which may have an important role in cancer outcomes [94]. Local anesthetics can modulate autonomic receptors, have a cytotoxic activity, and have an anti-inflammatory effect and anti-proliferative effects on mesenchymal stem cells [38,66]. However, whether the findings of preclinical research can have a clinical value depends on plasmatic concentration reachable after administration of anesthetics at human dose range. For example, the antitumor growth effect of ropivacaine and bupivacaine, two local anesthetics, occur at high concentrations and they do not affect apoptosis rate of pancreatic cancer cells at clinical concentrations in vitro [95]. To date only a retrospective study has reported that intraoperative intravenous lidocaine afforded a longer overall survival (HR = 0.616; 95% CI, 0.290–0.783; *p* = 0.013) in patients undergoing pancreatic cancer resection [96].

In the meantime, several randomized trials are being conducted on different cancer types (Table 1). The question is not simple, due also to the multifactorial and very complex situation including surgical, anesthetic, disease-specific and patients-linked factors. In this regard, the study of the influence of anesthesia on pancreatic cancer is at a very early phase of study and is based on retrospective studies.

Propofol afforded a better survival pattern than desflurane in pancreatic cancer surgery in a retrospective cohort study [97]. In this study, the overall mortality rate was 60.0% with propofol and 82.0% with desflurane (*p* = 0.006). Cancer-specific mortality rate was 57.0% with propofol and 78.0% with desflurane (*p* = 0.014) (HR 0.63) (95% CI, 0.42–0.95; *p* = 0.028). Recurrence was 43.0% with propofol and 66.0% with desflurane (*p* = 0.010) (HR 0.53 (95% CI, 0.34–0.84; *p* = 0.007) with no difference in postoperative metastases. Propensity score-matched HR for mortality and recurrence were respectively 0.63 (95% CI, 0.40–0.97; *p* = 0.037) and 0.55 (95% CI, 0.34–0.90; *p* = 0.028). Survival was 40.0% with propofol but 18% with desflurane (crude hazard ratio (HR 0.63 (95% confidence interval (CI), 0.42–0.93; *p* = 0.021) and HR, 0.53; 95% CI, 0.32–0.86; *p* = 0.010 in the multivariable analysis after adjustment). All cause survival was better with propofol than with desflurane (HR 0.63 (95% CI, 0.42–0.93; *p* = 0.021), propensity score matched HR 0.65 (95% CI, 0.42–0.99; *p* = 0.047). Conversely, desflurane anesthesia increased all-cause mortality, cancer-specific mortality and cancer progression [97].

Perioperative epidural anesthesia and analgesia and regional anesthesia have been evaluated for cancer survival after pancreatic surgery [98]. Epidural and regional anesthesia block nociception and reduce adrenergic activity, thus favoring the parasympathetic system [1]. However to date results of clinical trials are negative. In a retrospective study on 366 patients, epidural-TIVA (propofol-lidocaine-dexmedetomidine-ketamine) showed no differences in recurrence, metastasis, mortality, platelet-lymphocyte ratios (PLR) and neutrophil-lymphocyte ratios (NLR), compared with epidural-volatile-opioid anesthesia [midazolam-propofol-fentanyl-desflurane-hydromorphone] in pancreatic cancer surgery [99]. Recently, a heterogeneous meta-analysis including three RCT revealed no difference between regional and general anesthesia for late stage abdominal cancers [100].

Perineural invasion is a common feature of PC and is driven by the adrenergic system through neurotrophins. The sympathetic system acts though beta-adrenergic receptors (β-ARs) to favor PC activity. Conversely, beta-blocking agents such as propranolol inhibit PC progression [1]. To date three trials are evaluating the role of propranolol as an adjuvant to other anticancer treatments for PC (Table 1).

Finally, on the basis of retrospective studies, dexamethasone associated to peridural anesthesia may be associated with a reduced mortality risk and an improved recurrence-free after pancreatic cancer surgery [97,98].

## 5. Conclusions

Based on preclinical observations, it is to be hoped that the epigenetic effects of anesthetics on the biological mechanisms that determine the natural history of cancers in the perioperative period will be better investigated. Previous preclinical studies show that anesthetics induce epigenetic changes that can last for months and can be intergenerational [18]. This is an important observation, since such an epigenetic effect may be possible in residual cancer disease after surgery. Moreover, anesthesia alters host response and epigenetics of surgical tissues [35,36,37,38]. Most studies have investigated RNAs and there are only few studies investigating the other epigenetic mechanisms. The ability of anesthetics to affect epigenetics in germinal cells can be also a plausible sign that immature cancer cells may be susceptible to anesthetics.

Most preclinical or clinical studies still address single gene products and results are difficult to interpret in a setting where thousands of other genes interact with each other, along with other perioperative factors. One of the most investigated anesthetics is propofol and its role in the modulation of miRNAs. There is thus a need for studies investigating the whole genome. Microarrays, new generation sequencing (NGS) and RNAseq techniques can capture all the RNA-related information in a single test. Their integration into clinical trials may uncover unanticipated findings whose clinical impact is, at present, impossible to quantify. Several ‘proof-of-principle’ works show that gene expression profiling can predict important clinical outcomes and therefore have the potential to evolve into true diagnostic tests or drive clinical decision and implement personalized clinical care [89,101,102,103]. To date, there are no available studies of genome-wide analysis of the expression changes in tumors and surgical tissues induced by different anesthesia techniques.

Secondly, there is a need for robust prospective clinical studies, exploring anesthetics (propofol, lidocaine, nonsteroidal anti-inflammatory drugs, ketamine and propranolol), according to standardized outcomes, stratified in order to detect differences in the type and stage of cancer, and integrating genetic characterization of cancers [89,104,105]. The Standardizing Endpoints for Perioperative Medicine Group used a Delphi process to recommend, as appropriate endpoints for prospective trials of anesthetic technique on cancer outcomes, the cancer health related quality of life, days alive and out of hospital at 90 days, time to cancer progression, disease-free survival, cancer-specific survival, and overall survival (and 5-yr overall survival) [105]. The hypothesis that anesthetics may play a role in the recurrence of neuroendocrine pancreatic cancers and after endoscopic ultrasound-guided tumor ablation for pancreatic cancer should also be investigated [106,107].

Genetic profiling is a fundamental tool in order to take the disease into account in terms of clinical and genetic variability [89]. There is thus a need for randomized trials comparing different anesthetic techniques that integrate genetic profiling and genome-wide analysis of the gene expression changes [103,108]. A prospective trial design for genomic studies minimizes experimental bias and ensures that all samples are handled and processed in a standardized fashion [101,109]. To date only one study protocol (NCT03779685: GENe EXpression After Regional or General ANesthesia in Patients Undergoing Breast Cancer Surgery (GENEXAN)) has been built to incorporate gene expression analysis into a randomized trial comparing regional to general anesthesia for breast cancer surgery.

Rather than a single disease, cancer is an extremely variable disease from a genetic point of view. The higher the variability of the disease or population, the more the risk of losing effects on subpopulations in large trials who do not show variability. The MINDACT trial, for example, stratified breast cancer patients according to clinical and genetic risk instead of clinical risk alone, finding that patients with a high clinical risk but with a low genomic risk, randomized to chemotherapy or not, showed only a 2.6 percentage points benefit on the treatment group at 8 years follow-up, signifying that nearly 46% of high clinical risk patients can be treated with endocrine therapy alone [89]. A crucial aspect of generalizing clinical trials is that, in complex situations (cancer’s multiple phenotype, a long disease history, lack of a receptorial mechanism), we cannot expect a linear relationship between the action (anesthesia) and the outcome.

Another problem that may cover the effect of anesthesia and reduce the enthusiasm for further investigation is surgical curability, which is high today, especially in low grade cancers [3]. On the other hand the main cause of postoperative cancer death is metastasis, which occurs in one third of operated patients [110,111,112]. Only a small number of patients with pancreatic cancer will be operable and, among these, 7% will survive at 5 years, indicating a high recurrence rate and a lack of effective treatments [113]. It is well known that epigenetic mechanisms drive cancero-genesis in pancreatic cancer [114]. For these reasons, studies on epigenetics and pancreatic cancer with different anesthetics can be a promising field of research.

## Figures and Tables

**Figure 1 medicina-58-01102-f001:**
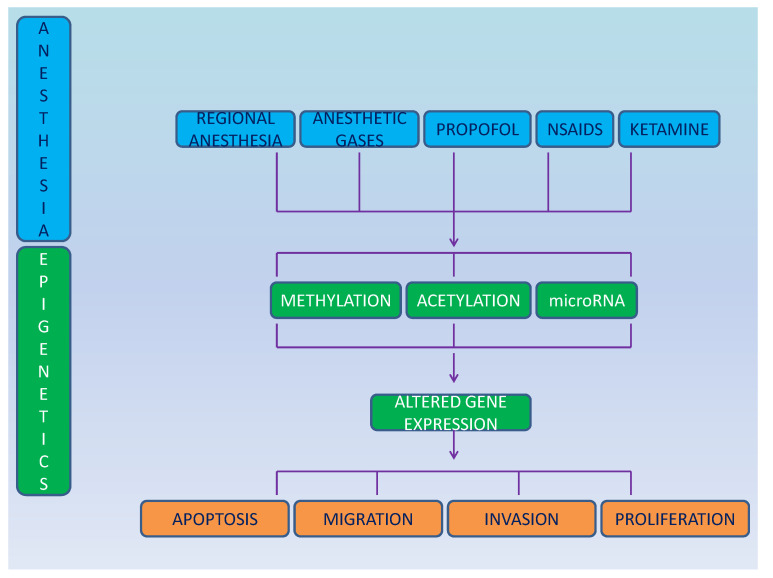
Schematic representation of the effect of anesthesia on epigenetics and cancer cells.

**Table 1 medicina-58-01102-t001:** Ongoing trials on pancreatic cancer and recurrence-related outcomes with repurposed anesthetics.

Identifier	Status	Intervention	Type	Condition	Outcome	Title
NCT02335151	C	Desflurane Propofol	Randomized Parallel assignment Double blind	86 patients Adenocarcinoma Circulating Tumor Cells Pancreatic Neoplasms Surgery	Peak of CTC in the postoperative phase after curative tumor removal Kinetics of CTC after surgery up to day 7 Month to Tumor recurrence Number of surviving patients	CTC Pancreatic Adenocarcinoma
NCT04048278	R	Intravenous (IV) lidocaine Saline Solution for Injection	Randomized Parallel Assignment Double blind	46 patients Pancreatic Cancer Surgery	Enzymatic activity of SCR TK on CTC Cytokine levels Chemokine levels Gene expression CTC number	Lidocaine Infusion in Pancreatic Cancer
NCT04449289	NYR	Intravenous (IV) lidocaine Epidural ropivacaine	Randomized Parallel Assignment Open label	100 patients Pancreatic Cancer Surgery	1- and 3-years recurrence rate after surgery Lidocaine and ropivacaine concentration Complication rate after surgery	Influence of Local Anesthetic Administration on the Cancer Recurrence Rate After Pancreatic Oncologic Surgery
NCT03245346	R	GEA PCEA GA PCIA	Randomized Parallel Assignment Open label	540 patients Cancer of Pancreas Surgery	Overall survival (OS) Disease-free survival (DFS) Postoperative pain score and side effects of patient-controlled Incidence of delirium Incidence of persistent post-surgical pain (PPSP) after surgery Length of stay in hospital after surgery and total costs after surgery Return of bowel function Start of enteral tube feeding Removal of Perianastomotic drains Removal of Urinary drainage Removal of nasogastric tube Removal of enteral feeding tube Blood level of neuroendocrine, stress and inflammatory response (blood epinephrine, norepinephrine, cortisol, VEGF, interleukin-6 (IL-6), interleukin-8 (IL-8), peripheral blood NLR (neutrophil-lymphocyte ratio)) Serum CA19-9, CA125, CEA, CA72-4, CA242, AFP, CA15-3, CA50 levels Plasma levels of ropivacaine and sufentanil	Effects of Epidural Anesthesia and Analgesia on the Prognosis in Patients Undergoing Pancreatic Cancer Surgery
NCT04025840	R	Epidural Block Dexamethasone	Randomized Intervention Factorial Assignment Intervention Model Single blind	260 patients Pancreatic Cancer Surgery	2-year overall survival Postoperative gastrointestinal complications Overall postoperative complications Length of stay in hospital after surgery All-cause 30-day mortality Quality of life in 1- and 2-year survivors Hospital readmission within 2 years after surgery 2-year progression-free survival Subjective sleep quality: Pain severity Time to ambulation after surgery Time to oral intake after surgery	Perioperative Epidural Block and Dexamethasone in Pancreatic Cancer Surgery
NCT03447691	R	Desflurane Total Intravenous Anesthesia with Propofol and Remifentanil	Randomized Parallel Assignment Triple blind	132 patients Pancreatic Cancer Distal CBD Cancer Surgery	Score of QoR40 Score of QoR40	Comparison Between Volatile Anesthetic-desflurane and Total Intravenous Anesthesia With Propofol and Remifentanil on Early Recovery Quality and Long Term Prognosis of Patients Undergoing Pancreatic Cancer and Common Bile Duct Cancer Surgery
NCT03838029	R	Propranolol Etodolac Other: Placebo	Randomized Intervention Parallel Assignment Quadruple blind	210 patients Pancreatic Neoplasms Surgery	Rate of cancer recurrence Biomarkers in extracted tumor tissue samples Number of patients with treatment related adverse events Depression, Anxiety, Global distress Fatigue	Perioperative Intervention to Reduce Metastatic Processes in Pancreatic Cancer Patients Undergoing Curative Surgery (BC-PC)
NCT03034096	R	Propofol Volatile Agent (sevoflurane, isoflurane, or desflurane)	Randomized Parallel assignment Double blind	2000 patients Patients with known or suspected cancer and scheduled to undergo any of the following oncologic surgical procedures: Lobectomy or pneumonectomy Esophagectomy Radical (total) cystectomy Pancreatectomy Partial hepatectomy Hyperthermic intraperitoneal chemotherapy (HIPEC) Gastrectomy (subtotal or total) Cholecystectomy or bile duct resection	All-cause mortality (Time Frame: 2 year minimum) Time to event Secondary Outcome Measures: Recurrence free survival (Time Frame: Minimum 2 years) Time to event All-cause mortality as a binary outcome (Time Frame: 2 years)	General Anesthetics in CAncer REsection Surgery (GA-CARES) Trial (GA-CARES)
NCT02660411	C	Sevoflurane Propofol	Randomized Parallel Assignment Triple blind	1228 patients Age ≥ 65 years and < 90 years; Primary malignant tumor; Do not receive radiation therapy or chemotherapy before surgery; Scheduled to undergo surgery for the treatment of tumors, with an expected duration of 2 h or more, under general anesthesia;	Over survival after surgery. (Time Frame: Up to 5 years after surgery.) Time from surgery to the date of all-cause death. Secondary Outcome Measures: Recurrence-free survival after surgery (Time Frame: Up to 5 years after surgery) Time from surgery to the date of cancer recurrence/metastasis or all-cause death, whichever occurs first. Event-free survival after surgery (Time Frame: Up to 5 years after surgery) Time from surgery to the date of cancer recurrence/metastasis, new cancer, new serious non-cancer disease, or all-cause death, whichever occurs first. Quality of life in 3-year survivors after surgery. (Time Frame: Assessed at the end of the 3rd year after surgery.) Quality of life is assessed with the European Organization for Research and Treatment of Cancer Quality of Life Questionnaire (EORTC QLQ-C30). Cognitive function in 3-year survivors after surgery. (Time Frame: Assessed at the end of the 3rd year after surgery.) Cognitive function is assessed with the Telephone Interview for Cognitive Status-Modified (TICS-m).	Impact of Inhalational Versus Intravenous Anesthesia Maintenance Methods on Long-term Survival in Elderly Patients After Cancer Surgery: a Randomized Controlled Trial
NCT03838029	R	Propranolol Etodolac Placebo	Randomized Parallel Assignment Quadruple blind	210 patients Pancreatic Neoplasms	Primary: -Rate of cancer recurrence (Time Frame: From the date of surgery until malignant disease is identified, assessed up to 60 months post-surgery) -Data regarding post-surgical recurrence will be recorded at 1, 3, 6, 12, 18, 24, 36, 48, and 60 following surgery -Biomarkers in extracted tumor tissue samples (Time Frame: An average of one year following surgery) Epithelial-to-mesenchymal-transition (EMT) status and natural-killer cell, macrophage, T-cell, and B-cell infiltration levels into tumor tissue (as assessed by messenger RNA profiling of tissue samples. -Biomarkers in blood samples (Time Frame: An average of one year following surgery) Cytokine levels in blood samples (interleukin-6, interleukin-10, C-reactive protein, interferon-gamma, and vascular endothelial growth factor and additional exploratory analysis of other cytokines)	Perioperative Intervention to Reduce Metastatic Processes in Pancreatic Cancer Patients Undergoing Curative Surgery
NCT05451043	NYR	Biological: Durvalumab Drug: Gemcitabine Drug: Nab paclitaxel Biological: Tremelimumab Drug: Propranolol Drug: Cisplatin	Single group Single Group Assignment Open label	62 patients Pancreatic Cancer Hepatocellular Cancer Biliary Tract Cancer Cholangiocarcinoma	Primary: -Investigating and establishing the efficacy of propranolol in boosting the effects of immunotherapy in pancreatic adenocarcinoma (Time Frame: Assessed one year after enrollment of last participant) combination of gemcitabine+nab-paclitaxel+propranolol+durvalumab+tremelimumab’s objective response rate is greater than or equal to 50% -Investigating and establishing the efficacy of propranolol in boosting the effects of immunotherapy in hepatocellular carcinoma (Time Frame: Assessed one year after enrollment of last participant) propranolol + durvalumab + tremelimumab objective response rate is greater than 45% Investigating and establishing the efficacy of propranolol in boosting the effects of immunotherapy in biliary tract tumors (Time Frame: Assessed one year after enrollment of last participant) -To demonstrate in unresectable BTC (gallbladder, cholangiocarcinoma of the biliary tracts including ampullary carcinomas) that the combination of gemcitabine + cisplatin + propranolol + durvalumab + tremelimumab’s objective response rate is greater than 50%.	Durvalumab and Tremelimumab in Combination With Propranolol and Chemotherapy for Treatment of Advanced Hepato-pancreabiliary Tumors (BLOCKED)
EudraCT number: 2018-000415-25		Propranolol Etodolac Placebo	Randomized Parallel Assignment Single blind	100 patients Non-metastatic head PC of the head undergoing elective pancreatoduodenectomy.	Primary: serious adverse events and reactions within 3 months. Secondary: utility of the two drugs in improving survival.	Pancreatic resection with perioperative drug repurposing of propranolol and etodolac: trial protocol of the phase-II randomized placebo controlled PROSPER trial.

R: recruiting; NYR: non yet recruiting; C: completed; CTC: Circulating Tumor Cells.

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
