# Peer review of "Anesthetics and Long Term Cancer Outcomes: May Epigenetics Be the Key for Pancreatic Cancer?"

_medicina, 2022, doi:10.3390/medicina58081102_

Round 1

Reviewer 1 Report

The paper is OK and i have only a single comment. Authors comment widely about anesthetics during surgical procedures, but some comments on anesthetics during endoscopic procedures should be added as well. In particular, comment on endoscopic ultrasound-guided tumor ablation or celiac plexus neurolysis in patients with pancreatic cancer, citing the paper PMID: 27356212

I would add a comment also on the role of anesthetics in patients undergoing surgery for pancreatic neuroendocrine tumors metastatic to the liver, citing the paper PMID: 27956320

Author Response

Dear reviewer

Thank you for your comments. Endoscopic procedures do not have the same impact on stress response as surgical procedures do. Moreover, to our knowledge, there is no literature on this. 

As we state in the paper, knowledge regarding anesthetic action on pacreatic cancer including neuroendocrine types is limited. 

Reviewer 2 Report

1. There is no chapter on pancreatic cancer, especially in terms of patophysiology.

2. The review work should contain figures - pleasae include them in individual chapters.

Author Response

Dear reviewer,

Thank you a lot for you comments. 

We did not include a chapter on pancreatic cancer pathofhysiology for shortness and because there are already full of reviews in this sense. We aimed to show what is the state of knowledge and what can be a novelty. 

Regarding figures, we will need more time and we ask the editor's confirmation. However we do not think it is essential to the scope of the paper. 

Round 2

Reviewer 1 Report

None of the points raised by myself and the other reviewer were addressed by the authors. Therefore, i don't consider the current version of the manuscript suitable to be accepted.

Author Response

Dear reviewer. Please fidn changes in the attached files. 

Sincerely

ZM

Reviewer 2 Report

The review work should contain figures - pleasae include them in individual chapters.

Author Response

(The authors gave the same response as above.)

Round 3

Reviewer 1 Report

The authors state that both of my previous points were addressed but in the current version of the paper i see only the second point was actually satisfied whereas the first point was not considered at all. 

Author Response

1

The authors state that both of my previous points were addressed but in the current version of the paper i see only the second point was actually satisfied whereas the first point was not considered at all.

Dear reviewer,

Thank you for your comments. We had already included your suggestions in the Conclusion chapter (in red) and cited your suggested articles. Please read the sentence: “The hypothesis that anesthetics may play a role in the recurrence of neuroendocrine pancreatic cancers and after endoscopic ultrasound-guided tumor ablation for pancreatic cancer should alsomay be investigated [106, 107]. Endoscopic procedures carry a low surgical stress risk, and for this reason the effect of anesthetics may be stand out better.”

As you may agree, clinical literature regarding this issue is scarce or even absent when considering epigenetic changes. For these reasons at the moment we can only build some hypotheses. Celiac plexus neurolysis concerns palliative care and for this reason we have not included this item in the discussion.

Sincerely

ZM